# Can ozone be used to calibrate aerosol photoacoustic spectrometers?

Donald A. Fischer[1,2] and Geoffrey D. Smith[1]

[1] Department of Chemistry, University of Georgia, 140 Cedar Street, Athens, Georgia 30602
[2] *Current Address:* Department of Chemistry and Physics, Western Carolina University, 111 Memorial Drive, Cullowhee, NC 28723

**Correspondence:** Geoffrey D. Smith (gsmith@chem.uga.edu)

**Abstract.** Photoacoustic spectroscopy (PAS) has become a popular technique for measuring absorption of light by atmospheric aerosols in both the laboratory and in field campaigns. It has low detection limits, measures suspended aerosols, and is insensitive to scattering. But PAS requires rigorous calibration to be applied quantitatively. Often, a PAS instrument is either filled with a gas of known concentration and absorption cross section, such that the absorption in the cell can be calculated from the product of the two, or the absorption is measured independently with a technique such as cavity ringdown spectroscopy. Then, the PAS signal can be regressed upon the known absorption to determine a calibration slope that reflects the sensitivity constant of the cell and microphone. Ozone has been used for calibrating PAS instruments due to its well-known UV-visible absorption spectrum and the ease with which it can be generated. However, it is known to photodissociate up to approximately 1120 nm via the $O_3 + h\nu\ (> 1.1\text{eV}) \rightarrow O_2(^3\Sigma_g^-) + O(^3P)$ pathway, which is likely to lead to inaccuracies in aerosol measurements. Two recent studies have investigated the use of $O_3$ for PAS calibration but have reached seemingly contradictory conclusions with one finding that it results in a sensitivity that is a factor of two low and the other concluding that it is accurate. The present work is meant to add to this discussion by exploring the extent to which $O_3$ photodissociates in the PAS cell and the role that the identity of the bath gas plays in determining the PAS sensitivity. We find a 5% loss in PAS signal attributable to photodissociation at 532 nm in $N_2$ but no loss in a 5% mixture of $O_2$ in $N_2$. Furthermore, we discovered a dramatic increase of more than a factor of two in the PAS sensitivity as we increased the $O_2$ fraction in the bath gas, which reached an asymptote near 100% $O_2$ that nearly matched the sensitivity measured with both $NO_2$ and nigrosin particles. We interpret this dependence with a kinetic model that suggests the reason for the observed results is a more efficient transfer of energy from excited $O_3$ to $O_2$ than to $N_2$ by a factor of 22-55 depending on excitation wavelength. Notably, the two prior studies on this topic used different bath gas compositions, and although the results presented here do not fully resolve the differences in their results they may at least partially explain them.

## 1 Introduction

Photoacoustic spectroscopy (PAS) has become a popular technique for measuring absorption of light by atmospheric aerosols (e.g. Roessler and Faxvog (1980); Japar and Szkarlat (1980); Moosmüller et al. (1998); Arnott et al. (1999); Lewis et al. (2008); Lambe et al. (2013); Wiegand et al. (2014); Zhang et al. (2016), among others). It is a desirable method because it has low detection limits, is capable of measuring suspended aerosols, and is insensitive to scattering. However, PAS requires rigorous

calibration for accurate absorption measurements, and this calibration becomes more difficult as the complexity of the PAS increases (e.g. with a multipass enhancement cell in which the sample interacts with multiple reflections of the excitation laser beam and/or the use of multiple wavelengths). Although ozone has been used as a calibrant for PAS (Lack et al., 2006, 2012), recent works exploring its validity at visible wavelengths have come to contradictory conclusions: Bluvshtein et al. (2017) saw a discrepancy between ozone calibrations and particle-based calibrations at 405 nm, while Davies et al. (2018) found this not to be the case. Concurrent to these publications, we have been exploring the use of ozone as a PAS calibrant for multipass, multi-wavelength, aerosol photoacoustic spectrometers; our observations are presented here to add to the discussion on the topic.

An underlying assumption of PAS is that energy imparted toward the electronic excitation of the analyte is quickly and efficiently transferred to translation energy in the bath gas molecules and does not contribute to non-thermal modes of relaxation such as luminesence or photochemistry. (Harshbarger and Robin, 1973) When the light is modulated on and off at acoustic frequencies, a pressure wave is produced that is detectable by a microphone. (Miklós et al., 2001) However, for quantitative measurements, this requires that no non-thermal relaxation pathways (e.g. photodissociation, fluorescence) exist, as any energy transferred non-thermally does not contribute to the PAS signal. Further, for trace gases in a bath gas, the excited analyte molecule must efficiently transfer its energy to the bath gas, and the bath gas must relax more quickly than the modulation frequency of the PAS. For accurate PAS measurements, the sound intensity (volume) measured with the microphone must be calibrated to units of absorption. For consistency, we will refer to this value as the sensitivity factor, $m$ with units of $(V/W)/Mm^{-1}$:

$$m = \frac{s}{b_{abs}} \tag{1}$$

where $s$ is the power-normalized PAS signal (i.e. in units of V/W) and $b_{abs}$ is the corresponding known absorption due to a calibrant (in units of $Mm^{-1}$). Most commonly, $m$ is determined either by filling the sample cell with a gas of known concentration ($N$) and absorption cross section ($\sigma$) (such that $b_{abs} = N\sigma$) or measuring the absorption with another technique such as cavity ringdown spectroscopy. By using multiple concentrations (or sizes, in the case of aerosols), a linear regression of $s$ vs. $b_{abs}$ can be performed from which the slope, $m$, can be determined. Examples of calibrants include aerosol particles such as flame-generated soot (Arnott et al., 2000) and gas-phase absorbers such as ozone (Lack et al., 2006, 2012) or nitrogen dioxide (Arnott et al., 2000; Lewis et al., 2008; Cross et al., 2010). Although ozone absorbs weakly in the UV-A and violet regions of the spectrum and is difficult to measure at those wavelengths, it has been employed for field calibrations because it can be easily generated using a UV lamp or corona discharge.

As noted above, Bluvshtein et al. (2017) conducted a systematic study of calibrants for a multipass photoacoustic spectrometer. They measured size-selected, light-absorbing aerosols, including nigrosin, Suwannee River fulvic acid, and Pahokee peat fulvic acid. They then used an independently measured refractive index (for nigrosin) or a refractive index determined from broadband extinction measurements (for SRFA and PPFA) and Mie theory to calculate the known absorption for each sample and found generally good agreement between their sensitivity factors; however, when they performed a calibration with ozone using a 405 nm laser, they found a much lower sensitivity factor (by roughly 50%). Alternatively, Davies et al. (2018) found

their measured nigrosin absorption cross sections agreed well with Mie theory at laser wavelengths of 405, 514 and 658 nm when they calibrated their PAS with ozone prior to nigrosin measurements. One difference between these two studies was the composition of the bath gas (sample matrix). The $O_3$ calibrations performed by Bluvshtein et al. were conducted in a bath gas composed of 90% $N_2$ and 10% $O_2$, while the calibrations of Davies et al. were performed in a bath gas of 75% $N_2$ and 25% $O_2$ (with an ozonated oxygen flow added to ambient air). If energy transfer from the excited state of ozone to the bath gas were different for these two systems, the effects may be easily explained; in fact, early PAS studies used the technique to measure relaxation rates of excited gas-phase molecules. (Harshbarger and Robin, 1973)

Clearly, there are contradictory results regarding the use of ozone as a calibrant for photoacoustic spectroscopy, and additional inquiry into the subject is warranted. Not discussed in either of the studies is a reason for the observed results. We note that ozone is well-known to photodissociate at wavelengths less than approximately 1120 nm, suggesting that PAS calibrations using ozone may be subject to non-thermal relaxation (Yung and DeMore, 1999). This could potentially explain discrepancies between ozone calibrations and other methods. In this communication, we attempt to provide some insight toward a more thorough understanding of this topic. Specifically, we compare calibrations with (1) $NO_2$, (2) nigrosin aerosols, and (3) ozone under various conditions. We find agreement between $NO_2$ and nigrosin but observe a lower sensitivity with ozone calibrations. We further show direct evidence for photodissociation of ozone inside the PAS when exposed to a 532 nm continuous wave laser and observed that adding small amounts (<5%) oxygen to the sample line changed the calibration slope significantly to bring it more in line with the other methods. We propose that the oxygen dependence can be explained by a simple kinetic model in which oxygen deactivates the excited ozone more efficiently than does nitrogen. While this doesn't fully explain the differences between Bluvshtein et al. (2017) and Davies et al. (2018), the overall trend in our data is consistent with the trend observed in these studies – that a lower concentration of oxygen in the bath gas leads to a lower PAS calibration slope.

## 2 Materials and Methods

### 2.1 Photoacoustic Spectrometer

The photoacoustic spectrometer used in this study has been described previously elsewhere (Fischer and Smith, 2018). Briefly, it is a single-cell, four-wavelength, laser PAS. Four diode lasers (406, 532, 662, and 780 nm) are combined into a single beam with dichroic mirrors and turned into a multipass cell consisting of two highly reflective, cylindrical mirrors (R > 99%); the front mirror has a 2 mm entrance hole drilled in the center (Silver, 2005). The PAS cell itself sits within the multipass cell and follows the design of Lack et al. (2006). A calibrated photodiode behind the rear multipass mirror is used to monitor the power of each laser simultaneously. The system includes a cavity ringdown cell (CRD) operating at 662 nm (from the same 662 nm laser employed by the PAS) for direct calibration of the PAS. The four lasers in the PAS are operated simultaneously at frequencies spaced every 2 Hz around the resonant frequency of the cell. A fast-Fourier transform (FFT) is performed on the microphone signal to deconvolve the signals at each wavelength. The resonant frequency of the PAS cell is measured by scanning the laser frequency across the resonant peak of the cell, typically filled with only the bath gas, and finding the best fit to the frequency sweep data. A frequency sweep was conducted prior to each set of measurements and anytime the gas type

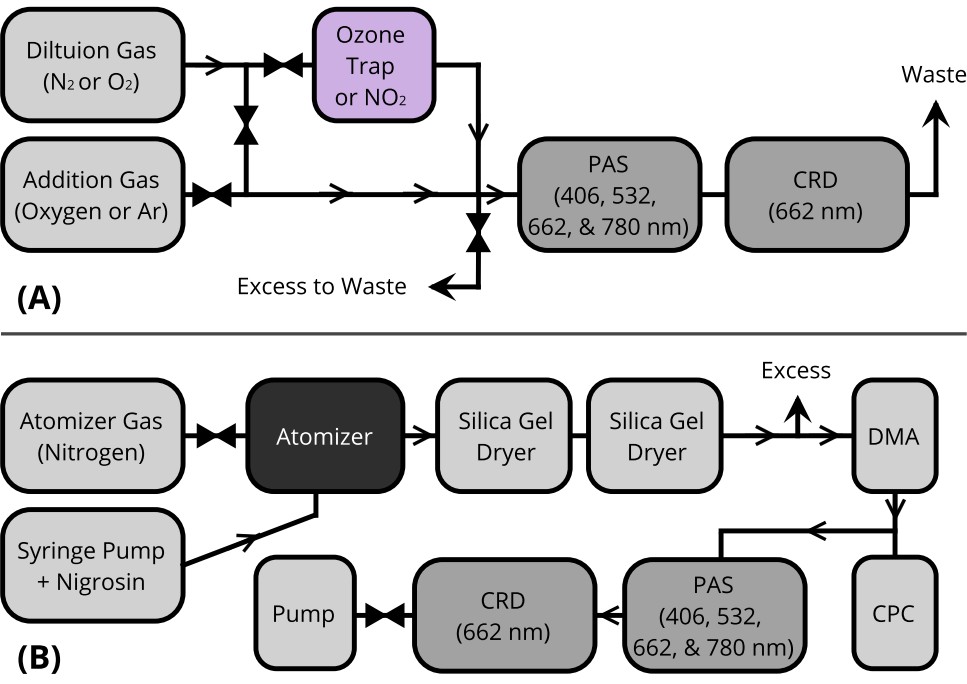

**Figure 1.** Block diagram of the experimental setup. (A) Setup used for $O_3$ and $NO_2$ measurements, and (B) setup used for nigrosin measurements. Triangles indicate mass flow controllers or critical orifices; arrows indicate direction of flow. CPC = condensation particle counter; DMA = differential mobility analyzer; CRD = cavity ringdown spectrometer.

was changed. From these sweeps, the quality factor, $Q$, of the cell was determined to be 30. The lasers can be individually switched from digital modulation (as is used for PAS) to continuous wave mode, which is helpful in conducting photolysis studies. The incident single-pass powers, which are representative of the powers experienced by each $O_3$ molecule, were 61, 32, 44, and 77 mW for 406, 532, 662, and 780 nm, respectively. A diagram and more thorough description of the instrument can be found in Fischer and Smith (2018).

## 2.2 NO$_2$ Measurements

Following our typical procedure, as described in Fischer and Smith (2018), we calibrated the PAS by pushing a mixture of nitrogen dioxide in nitrogen through the instrument. A standard 10.29 ppm ($\pm$ 5%) mixture of $NO_2$ in $N_2$ with a trace of $O_2$ for stability (Airgas, Athens, Georgia) was diluted to various concentration into $N_2$ boil off from a liquid nitrogen dewar (Airgas, Athens, Georgia). The rotameter was used to measure the flow of $NO_2$ while the $N_2$ flow was controlled with a needle valve at approximately 200 sccm (standard cubic centimeters per minute) and measured with an electronic flow meter (TSI, Shoreview, Minnesota). The flow rate through the instrument was the sum of the two flows, and ranged from 225–400 sccm depending on the $NO_2$ flow rate. $NO_2$ was introduced first to the PAS cell and then transported to the CRD via a short length (10

cm) of copper tubing. The outlet of the CRD was plugged and the gas was directed out of the purge inlets to avoid dead volume in the cell (no purge flow was used for $NO_2$ measurements). CRD and PAS measurements were conducted simultaneously at 662 nm, and all other lasers were turned off during $NO_2$ measurements. Figure 1A shows a block diagram of the setup used for $NO_2$ measurements. The outlet of the PAS cell was open to atmospheric pressure, and as such the pressure inside the cell was free to fluctuate with the local ambient pressure. Likewise, the temperature was free to fluctuate with ambient temperature, but was within in the range of $22 \pm 2\,°C$ for all experiments. Prior to all $NO_2$ experiments, 10 ppm $NO_2$ was flowed through the cell at 1-2 SLPM for several minutes to passivate all components of the system.

### 2.3 Ozone Measurements

Ozone was generated using a commercial corona discharge ozone generator (Pacific Ozone, Benicia, California) with high-purity $O_2$ (99.999%, Airgas, Athens, Georgia). The ozone was trapped on silica gel in a glass trap held in slurry of solid $CO_2$ and ethanol at -73 °C. Prior to trapping, the silica gel and trap were heated to 100 °C while being held under vacuum for at least 1 hour to remove contaminants. As with $NO_2$, no purge flow was used during $O_3$ measurements and the sample was pushed to the PAS first and transported to the CRD via a short length of copper tubing. The outlet of the CRD was plugged and the sample was directed out of the purge flow lines to minimize dead volume inside the cell. Figure 1A shows a block diagram of the setup used for ozone measurements. Mass flow controllers were used to control the ratio of oxygen to nitrogen (MKS Instruments). The outlet of the PAS cell was open to atmospheric pressure, and as such the pressure inside the cell was free to fluctuate with the local ambient pressure. Likewise, the temperature was free to fluctuate with ambient temperature, but was within in the range of $22 \pm 2$ °C for all experiments. Ozone calibrations were performed at 532, 662, and 780 nm; 406 nm measurements were not conducted because of a very low signal to noise ratio at that wavelength for the relatively low ozone concentrations used.

### 2.4 Nigrosin Measurements

Figure 1B shows a block diagram of the setup used for nigrosin measurements. Nigrosin aerosol was generated using a constant output atomizer (TSI 3076) with an aqueous solution of nigrosin (4 $g\,L^{-1}$, Sigma Aldrich Catalog Number 198285, CAS# 8005-03-6, LOT MKBG7493V) and dried using a series of two silica-gel diffusion dryers. The relative humidity was kept below 5% and monitored with an inline relative humidity probe (HMP110, Vaisala Corportation, Helsinki, Finland). Atomized, dried particles were size selected at electrical mobility diameters of 500, 550, 600, and 650 nm using an electrostatic classifier (TSI 3080) and differential mobility analyzer with a 10:1 sheath flow to sample flow ratio and an 0.071 mm diameter impactor orifice to provide a cut point of approximately 1100 nm and reduce transmission of doubly charged particles (DMA, TSI 3085). Monodisperse aerosols were split in parallel to a condensation particle counter (CPC, TSI 3775) and the photoacoustic cell and delivered to each instrument through conductive silicone tubing. After particles passed through the PAS, they entered the CRD cell, which had a purge flow of 60 SCCM $N_2$ (maintained by a critical orifice) over each mirror to prevent particle deposition. The aerosol sample was pulled through the instrument with a diaphragm pump (KNF Neuberger, Inc., Trenton, NJ) and the flow rate was maintained at 330 SCCM total flow with a critical orifice (Lenox Laser, Glen Arm, Maryland). All lasers

were operated simultaneously. The refractive index from Bluvshtein et al. (2017) was used to calculate nigrosin absorption cross sections using Mie theory assuming a geometric standard deviation of 1.05. Mie theory calculations were performed in MATLAB.

## 3    Results and Discussion

We chose to take an alternate approach to calibrating with ozone compared to prior studies (Bluvshtein et al., 2017; Davies et al., 2018). Instead of using the flow directly out of an ozone generator, we trapped ozone on a silica gel trap prior to analysis. This allows us to achieve lower overall oxygen concentrations than available with an ozone generator and more fully map out the behavior of ozone in the presence of oxygen. Further, while others have used single-wavelength PASs in parallel, we used a 4-wavelength, single-cell PAS. This gave us the opportunity to operate some lasers in continuous-wave mode and probe for signal loss due to photodissociation. The results presented here will be discussed first in terms of our typical calibrant ($NO_2$) and a particle-based calibration (nigrosin). We will then discuss the use of ozone in relation to those calibrants and finally end with a discussion of oxygen's effect on ozone signals in the PAS.

### 3.1    Non-ozone Methods of Calibration

We prefer to calibrate with $NO_2$ by measuring the PAS signal at 662 nm and comparing to the absorption measured by the CRD at 662 nm. Because each of the instruments is illuminated by the same laser, the uncertainty is determined only by the uncertainty of the CRD and the precision of the PAS; all uncertainties associated with flow measurement and absorption cross sections are irrelevant. Further, because all of our wavelengths are contained in a single cell, the power-normalized calibration at 662 nm can be applied to all wavelengths (including 406 nm, at which wavelength $NO_2$ photodissociates) (Wiegand et al., 2014; Fischer and Smith, 2018). This approach, however, adds some additional uncertainty from the measurement of the effective power of each wavelength.

Performing the calibration with $NO_2$ using the CRD to determine absorption yields a calibration slope of $m = 11.9 \times 10^{-4} (\text{V/W})/\text{Mm}^{-1}$. Because we used a 10 ppm calibrated mixture of $NO_2$, we were also able to derive independently a calibration slope using the calculated $NO_2$ absorption from the product of the concentration, $N$, and the absorption cross section, $\sigma$, as measured by Burrows et al. (1999). This slope of $m = 11.7 \times 10^{-4} (\text{V/W})/\text{Mm}^{-1}$ is within 1.5% of the CRD method despite the larger uncertainty due to uncertainties in flow measurements. With nigrosin, we obtain a slope of $m = 10.7 \times 10^{-4} (\text{V/W})/\text{Mm}^{-1}$, within 10% of the $NO_2$ calibrations. The calibration curves for these methods can be seen in Figure 2. Although the agreement here is not bad, there is some discrepancy between ozone and nigrosin. We speculate this is due to errors with the nigrosin calibration due to (e.g.) CPC counting errors (accuracy = 10%) and/or errors or lot-to-lot differences in the refractive index of nigrosin.

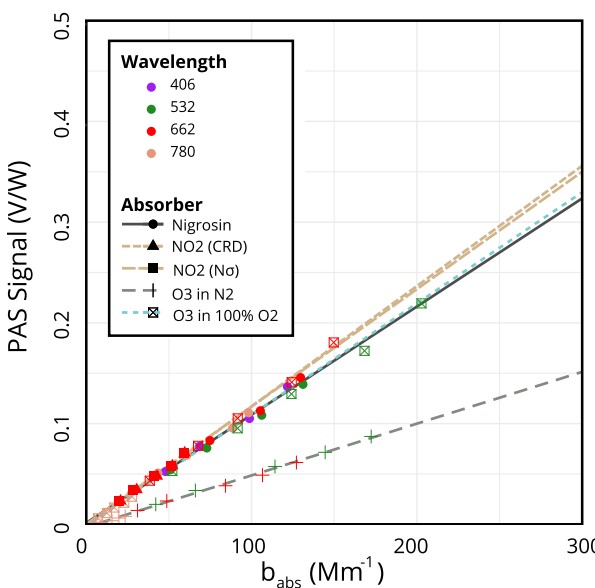

**Figure 2.** Calibration curves from various methods. Points are colored by wavelength.

### 3.2 Ozone as a Calibrant

We have observed discrepancies between ozone and $NO_2$ calibrations. In Figure 2, which shows the calibration data and fits to all wavelengths (for ozone that is 532, 662, and 780 nm), the most dramatic outlier is the dashed grey-green line obtained from ozone in pure $N_2$, which yields a slope more than 50% lower than the slopes obtained with $NO_2$ and nigrosin at $m = 5.1 \times 10^{-4} (V/W)/Mm^{-1}$. We hypothesized that this difference was due to photolysis of ozone under irradiation by visible light via $O_3 + h\nu \, (> 1.1eV) \rightarrow O_2(^3\Sigma_g^-) + O(^3P)$. (Burkholder et al., 2015) If this were the case, we further hypothesized that diluting ozone with oxygen instead of nitrogen would yield a larger calibration slope because the oxygen would promote recombination of $O(^3P)$ and $O_2$ to form ozone. Indeed, as the dotted teal line in Figure 2 indicates, the calibration slope fit to all three wavelengths under conditions of 100% $O_2$, $m = 9.8 \times 10^{-4}$ (V/W)/$Mm^{-1}$, was much closer to the slopes obtained using $NO_2$ or nigrosin. The slopes derived from fits to the data of the individual wavelengths are similar, as expected, since the calibration should be independent of the wavelength of light: $m = 9.6, 10.3$ and $9.6 \times 10^{-4}$ (V/W)/$Mm^{-1}$ for 532, 662 and 780 nm, respectively.

To search for evidence of $O_3$ photolysis, we operated our 532 nm laser in continuous-wave mode. This mode prevented the laser from contributing to the PAS signal and yielded maximum continuous power available for photodissociation. The PAS signal due to ozone was monitored with the 662 nm PAS channel, and the concentration of ozone was monitored with the cavity ringdown spectrometer using the absorption cross section of Burrows et al. (1999). This approach allowed us to

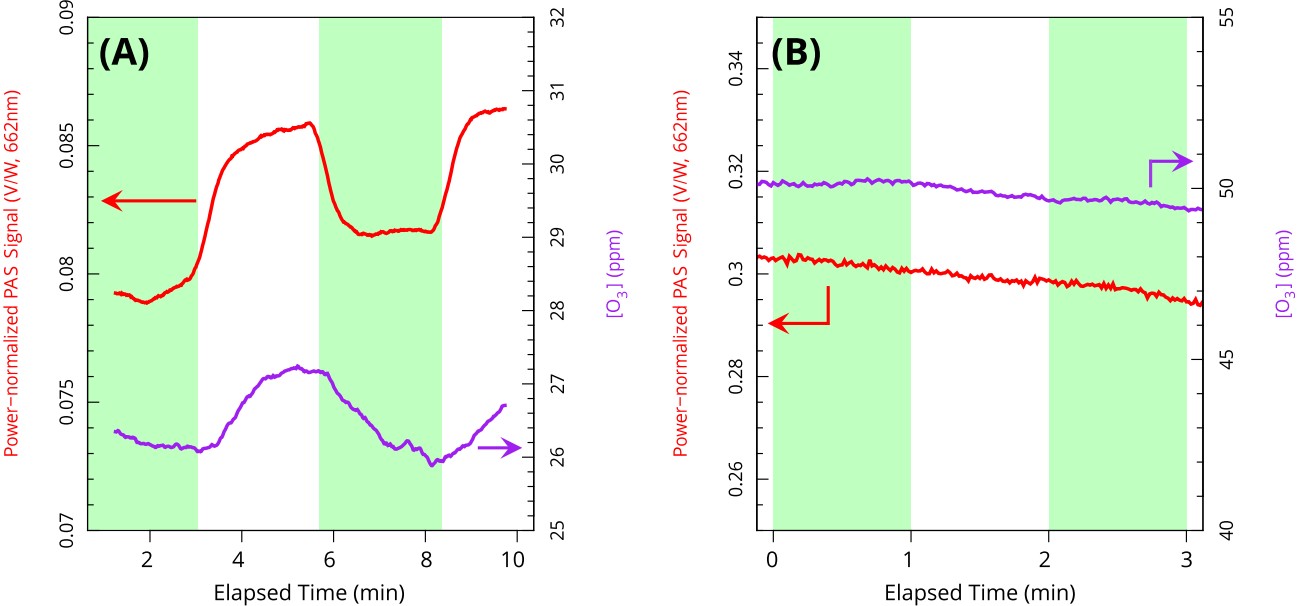

**Figure 3.** Photolysis of ozone in the PAS. (A) Response of 662 nm PAS signal and [$O_3$] as measured with the 662 nm CRD to irradiation at 532 nm with no oxygen present and (B) with 5% oxygen present. Green shaded regions represent times when the 532 nm laser was turned on and white regions when it was off. The slight downward drift evident is likely from a decreasing $O_3$ concentration as the trap becomes depleted.

separate effects due to a lowering of the ozone concentration (which would be evident with the CRD) and any additional loss of PAS signal resulting from energy loss due to photodissociation. The green shaded regions in Figure 3 indicate when the 532 nm laser was turned on to illuminate the ozone inside the PAS. An immediate decrease of 5% in both the PAS signal and the ozone concentration measured with the CRD is noticed, consistent with a loss of ozone due to photodissociation. However, a simple photolysis calculation assuming a unit quantum yield for photodissociation indicates that nearly all of the $O_3$ (more than 99.9%) should photodissociate. Given the small 5% loss observed, we conclude that a trace of $O_2$ must have been present thereby promoting re-formation of $O_3$; indeed, we estimate that only 4 ppm of $O_2$, perhaps coming from the $O_3$ trap or just a tiny leak of ambient air, would be sufficient to compete with the photolysis loss. The origin of the slight upward drift apparent in the PAS signal is not known, but it may indicate a shift in cell resonant frequency or temperature; nonetheless, the observed 5% loss of signal is substantially smaller than the 50% reduction in sensitivity observed in Figure 2. Furthermore, the loss due to the 532 nm light was not observed when we added 5% oxygen (of the total sample flow), as shown in Figure 3B, suggesting that in the presence of oxygen ozone is rapidly reformed. But how much oxygen is sufficient to accurately perform a PAS calibration with ozone? For example, it can be convenient to calibrate in air (i.e. 20% oxygen, for example Davies et al. (2018)) but is there a sufficient amount of oxygen to ensure the full sensitivity of the PAS?

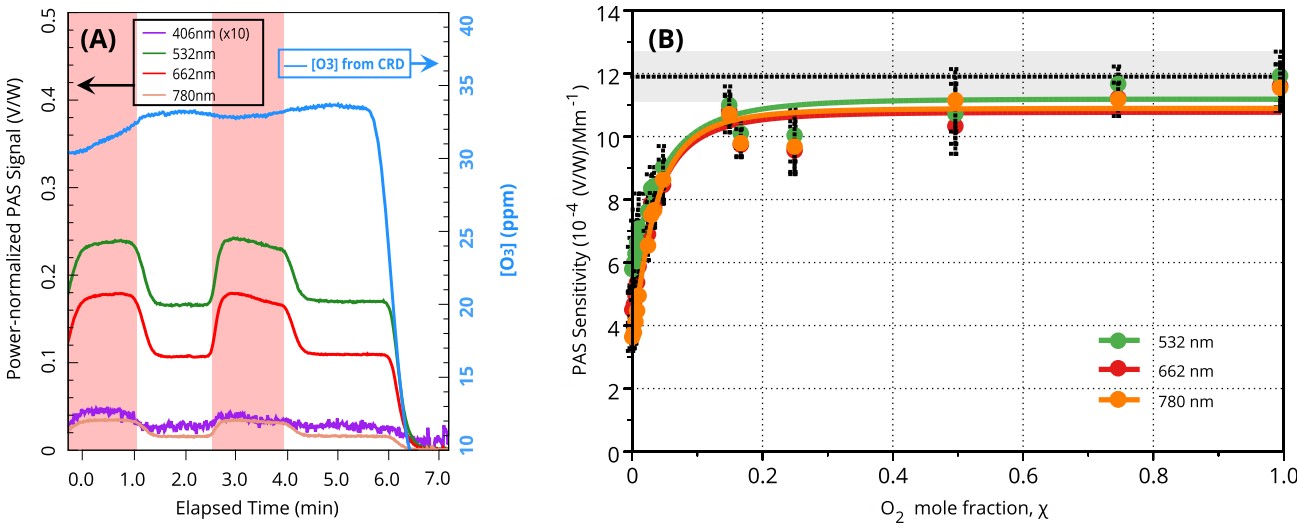

**Figure 4.** (A) Time series of $O_2$ addition; (B) PAS signal (normalized to absorption cross section) as a function of $O_2$ mole fraction, $\chi$. Red shaded regions in (A) represent times when 5% oxygen was added to the sample stream. Error bars in (B) are $\pm$ 1 SD of a 30-second average. The black line in (B) represents the value obtained with $NO_2$ with the gray shaded region representing the $3\sigma$ uncertainty, and the colored lines are the best fit to the data using Equation 3.

### 3.3 Effect of Oxygen on Ozone Signal

The effects of oxygen on the PAS signal can be seen clearly in Figure 4. In Figure 4A, oxygen was added to the sample line such that it made up 5% of the total flow. The red regions indicate when $O_2$ was added to the sample stream, and the white regions indicate when it was removed. There is a clear difference upon addition of $O_2$ to the sample flow, with it increasing the signal roughly 50-75%, and a similar trend was observed at all four measurement wavelengths available in our PAS. This effect cannot be due to changes in the concentration of ozone, which were monitored with the CRD and actually decreased slightly when oxygen was added (due to the slight dilution of the sample flow). An alternative explanation would be a shift in the resonant frequency upon addition of oxygen. However, because the resonant frequency was measured in nitrogen, any shift in resonant frequency should only decrease the signal. Further, measurements of the resonant frequency showed negligible differences between nitrogen-only samples and those with 5% oxygen added. Thus, the change in composition was not enough to have an appreciable effect on the resonant frequency of our low-$Q$ (wide-bandwidth) PAS cell. We therefore conclude that the observed increase in signal upon addition of oxygen is indeed attributable to a change in sensitivity accompanying the change in composition of the bath gas. Finally, such a phenomenon was not observed when adding argon instead of oxygen (data not shown), implying that the effect is attributable to the presence of oxygen specifically.

Figure 4B shows the effect of adding oxygen in varying amounts from 0–100% of the bath gas. A clear trend is observed in relation to the oxygen concentration at 532, 662, and 780 nm; the effect likely exists at 406 nm as well, but that wavelength

was not measured because of ozone's low absorption cross section at that wavelength. In the absence of $O_2$, the sensitivity is about $4 \times 10^{-4} (V/W)/Mm^{-1}$, more than a factor of 2 lower than the normal PAS cell sensitivity measured with either $NO_2$ or nigrosin. The sensitivity increases quickly as oxygen is added up to about 20% oxygen in nitrogen, at which point it begins to asymptotically approach an upper limit that is more in line with the sensitivities measured by other methods. Others

have observed similar effects measuring HCN when adding water vapor into the cell (Kosterev et al., 2006) and when adding oxygen into a mixture of $NO_2$ and $N_2$, although in that case adding oxygen caused a decrease in the signal (Kalkman and van Kesteren, 2008).

We note that the observed sensitivity dependence on bath gas composition could partially explain the lower sensitivity to $O_3$ compared to nigrosin particles observed by Bluvshtein et al. since the bath gas in that study contained only 10% $O_2$. Using

the data in Figure 4B, we estimate that the sensitivity would be 17% low, which is in the right direction but cannot explain the entire difference. Likewise, we estimate the sensitivity to $O_3$ in the work of Davies et al., which used 25% $O_2$ in the bath gas, to be 12% low. We conclude, then, that the different amounts of $O_2$ in the bath gas fo these two studies cannot fully explain the discrepancy between them.

An underlying assumption of PAS is that all the photon energy absorbed by the sample is transferred to the bath gas as

thermal energy to create an acoustic wave. This process requires efficient transfer of energy from the excited state of the analyte (e.g. $O_3{}^*$) into translational, rotational, and/or vibrational modes of the bath gas and the further relaxation of the bath gas molecule. However, if the transfer of energy from the analyte to the bath gas is inefficient or if the excited state of the bath gas, analyte, or another intermediate is long-lived with respect to the modulation frequency of the light source, the photon energy will not be efficiently converted to acoustic energy, which is observed as a decreased sensitivity. The observed

dependence on $O_2$ concentration indicates that energy transfer is more efficient with $O_2$ as the bath gas compared to $N_2$, and the shape of the dependence on $O_2$ concentration suggests a competitive kinetic model. Indeed, the data are fit reasonably well by a simple model in which $O_2$ and $N_2$ are each assumed to deactivate the $O_3{}^*$ in one step but with different rate constants, $k_{O_2}$ and $k_{N_2}$:

$$\frac{d[O_3{}^*]}{dt} = -k_{N_2}[O_3{}^*][N_2] - k_{O_2}[O_3{}^*][O_2] \tag{2}$$

Following the derivation of Kosterev et al. (2006), the sensitivity, $m$ (in $(V/W)/Mm^{-1}$), can be expressed as:

$$m = \frac{m_0}{\sqrt{1 + \left(\frac{A}{1+\frac{r\chi}{(1-\chi)}}\right)^2}} \tag{3}$$

where $m_0$ is the asymptotic sensitivity coefficient (i.e. with instantaneous relaxation), $\chi$ is the $O_2$ mole fraction, $r$ is the ratio of the quenching rate constants for oxygen and nitrogen ($k_{O_2}/k_{N_2}$), and:

$$A = 2\pi f \tau_{N_2} \tag{4}$$

where $f$ is the modulation frequency and $\tau_{N_2}$ is the deactivation lifetime of $O_3{}^*$ in 100% $N_2$ $(= \frac{1}{k_{N_2}[N_2]})$. For efficient conversion of the absorbed photon energy to acoustic energy, the deactivation rate must be significantly faster than the modulation frequency, meaning $A$ must be $<< 1$.

Fitting Equation 3 to each of the three data sets in Figure 4B results in reasonable fits with $R^2$ values of 0.96 or greater. The values of the $A$ parameter are 1.6, 2.2, and 3.0 for 532, 662, and 780 nm, respectively, reflecting the fact that the energy transfer in 100% $N_2$ is inefficient for all three wavelengths. The values of $r$, the ratio of the deactivation rate constants in $O_2$ and $N_2$, are 22, 37, and 55 for 532, 662, and 780 nm, respectively, reflecting the increased sensitivity in the presence of $O_2$. The differences in these values may reflect differences in the densities of states of the bath gas and the ozone when excited by the different wavelengths of light, though a more definitive interpretation is beyond the scope of this work. The values of $m_0$ are 11.2, 10.8, and $10.9 \times 10^{-4}$ (V/W)/Mm$^{-1}$ for 532, 662, and 780 nm, respectively, which indicate similar sensitivities in the limit of 100% $O_2$ for all three wavelengths and are within 10% of the sensitivity measured with $NO_2$. Clearly, however, the data appear not to have reached an asymptote even at 100% $O_2$, which may reflect the limitations of using such a simple model in which deactivation of $O_3^*$ by $N_2$ and $O_2$ is represented by single steps. Nonetheless, this model captures the general shape of the sensitivity dependence on $O_2$ concentration and provides a guide for assessing the relative efficiencies of the two bath gases. In fact, the measured values of the sensitivities at 100% $O_2$ are within 3% of the $NO_2$ measurement indicating that calibration with $O_3$ is a viable option as long as it is performed with 100% $O_2$ as the bath gas. It may even be possible to perform such a calibration with smaller concentrations of $O_2$ and use a correction based on a curve similar to that shown in Figure 4B, though the additional uncertainty incurred with doing so may make such an approach undesirable. Finally, we note that since the $A$ term is a function of $f$, the sensitivity of PAS measurements made at higher frequencies than that used here (1414 Hz in 100% $N_2$) will demonstrate an even more pronounced dependence on $O_2$ concentrations.

## 4    Conclusions

We show direct evidence of ozone photodissociation at 532 nm at the level of 5% inside a PAS cell. Despite the fact that this photodissociation pathway is well established, ozone has been used to calibrate aerosol PAS instruments with a dearth of discussion on the impact of photodissociation until very recently. Significantly, Davies et al. (2018) find good agreement between an ozone calibration and one performed with nigrosin particles, while Bluvshtein et al. (2017) measured an ozone calibration half that of the one obtained with nigrosin particles with no obvious explanation for the disparity. Here, we expand on this work by systematically investigating the dependence of the ozone sensitivity on $O_2$ concentration and performing kinetic modeling suggesting that $N_2$ as a bath gas results in inefficient deactivation of $O_3^*$. Interestingly, our results are not sufficient to entirely reconcile the differences between the findings of Bluvshtein et al. (2017) and Davies et al. (2018). In the prior, a bath gas composition of 10% $O_2$ and 90% $N_2$ was used, which would lead to a significantly lower (17%) calibration constant for ozone than for other calibrants but is insufficient to explain the factor of two discrepancy observed with nigrosin particles; in the latter, a composition of 25% $O_2$ and 75% $N_2$ was used, which would lead to a smaller (12%) discrepancy between ozone and nigrosin measurements. We find that ozone is a suitable calibrant for PAS in a bath gas of 100% $O_2$ but that its use at lower $O_2$ concentrations requires careful comparison to other calibrants, such as $NO_2$ or nigrosin particles, and will incur increased uncertainties associated with the necessary correction.

*Author contributions.* AF conducted the experiments; GS and AF composed the manuscript.

*Competing interests.* The authors declare no competing interests.

*Acknowledgements.* The authors thank Dr. Rawad Saleh of the University of Georgia College of Engineering for loaning the atomizer, syringe pump, and a diffusion dryer used in this study. They further appreciate the technical support provided by the University of Georgia Instrument Shop in machining of the PAS and CRD cells. Finally, they gratefully acknowledge the financial support of the National Science Foundation, Division of Atmospheric and Geospace Science (AGS-1241621 and AGS-1638307).

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
