# Peer review of "Can ozone be used to calibrate aerosol photoacoustic spectrometers?"

_Atmospheric Measurement Techniques, 2018_

## Referee Comment (RC1) · Anonymous Referee #1 · 10 Aug 2018

Review of 'Technical Note: Can ozone be used to calibrate aerosol photoacoustic spectrometers?', by Fischer and Smith, 2018.

**Summary and General Comments**

Photoacoustic spectrometers (PAS) require an accurate calibration for subsequent measurements of absorption. Fischer and Smith add to an ongoing discussion concerning the accuracy of PAS calibrations with ozone for applications pertaining to aerosol absorption measurements at laser wavelengths in the visible spectrum. To put this work in context, numerous researchers have used $O_3$ to calibrate PAS for aerosol absorption applications. However, Bluvshtein *et al*. reported a factor of two discrepancy between their $O_3$ calibration and that performed with an aerosol (nigrosin dye) calibrant at a PAS laser wavelength of 405 nm.[1] Meanwhile, a subsequent paper by Davies *et al*. reports excellent agreement between an $O_3$ calibration and that attained using nigrosin dye at multiple PAS laser wavelengths,[2] including the same 405 nm wavelength reported in the Bluvshtein *et al*. work. Bluvshtein *et al*. could not ascertain the source of their calibration discrepancy for using an $O_3$ calibrant and, as the current authors' state, further investigation into the accuracy of $O_3$ calibrations of PAS is warranted. Fischer and Smith first assess the impact of photodissociation at a 532 nm wavelength on the measured PAS response and then investigate how PAS sensitivity factor depends on the mole fraction of $O_2$ ($x_{O2}$) in the bath gas. In a pure $N_2$ bath gas, the PAS response is reduced by 5% when a continuous output 532 nm laser irradiates the gas sample simultaneous to a PAS modulation laser of 662 nm wavelength, while this loss in PAS response is negated by adding $O_2$ to the bath gas even at small $x_{O2} = 0.05$ concentrations. Notably, the PAS sensitivity factor is measured at 532, 662 and 780 nm with variation in the $O_2$ and $N_2$ mole fractions in the bath gas. The PAS sensitivity factor increases with $x_{O2}$, approaching a plateau at $x_{O2} > 0.2$ and is in near-exact agreement with an $NO_2$ and nigrosin aerosol calibration for $x_{O2} = 1$. However, the PAS sensitivity of the $O_3$ calibration for $x_{O2}$ close to zero is almost a factor of 3 smaller than that of the $NO_2$ calibration and is attributed to $N_2$ being very inefficient (compared to $O_2$) at converting the energy of excited $O_3$ to thermal energy through collisional relaxation. Ultimately, this work does not resolve fully the discrepancies between the Bluvshtein *et al*. and Davies *et al*. studies, but is a new and important contribution to the ongoing discussion of $O_3$-based PAS calibration, raising the important consideration of bath gas composition. Moreover, Atmospheric Measurement Techniques is a suitable location for this work. Therefore, I recommend the publication of this Technical Note after the following comments have been addressed.

**Specific comments**

Page 1 line 18: The authors highlight the work of Bluvshtein *et al*. and Davies *et al*. in the abstract as partly motivating the current study. Therefore, the authors should summarise how the current work may reconcile these two studies, if at all, at the end of the abstract. As discussed further below, the authors incorrectly state in their conclusions that both the Bluvshtein *et al*. and Davies *et al*. studies use a bath gas of 10% $O_2$ in $N_2$; while this is the case for the Bluvshtein study, the Davies study is for ozone injected into ambient air (25% $O_2$ in $N_2$). These key differences in bath gas composition may partly account for calibration discrepancies between the two studies.

Page 1 line 25: Some readers may not be familiar with a 'multipass enhancement cell'. The authors should make clear at this early stage what is meant by 'multipass' (i.e. multiple reflections of the excitation laser beam through the sample), although it is acknowledged that specific details of the multipass optical cell is given on page 3 line 11 – page 4 line 1.

Page 2 line 1: '…recent works exploring its validity have come to contradictory conclusions' should emphasise that these works were for visible laser wavelengths, i.e. '…recent works exploring its validity at visible wavelengths have come to contradictory conclusions'.

Page 2 line 2: '…discrepancy between ozone calibrations and particle-based calibrations'. The authors should state that this study was at a PAS laser wavelength of 405 nm only.

Page 2 line 16: 'Most commonly, this…'. 'This' is ambiguous and perhaps should read, 'Most commonly, this sensitivity factor…'.

Page 2 line 25: Replace '…light absorbing aerosols' with '…size selected light absorbing aerosols'.

Page 2 line 27 - 28: Replace 'when they performed a calibration with ozone' with 'when they performed a calibration with ozone at a laser wavelength of 405 nm'.

Page 2 line 29: Add after 'Mie theory', 'at laser wavelengths of 405, 514 and 658 nm'.

Page 2 line 32: The authors should state here: 'However, we note that the $O_3$ calibrations performed by Bluvshtein $et\ al.$ were in a bath gas composed of 90% $N_2$ and 10% $O_2$, while the calibrations of Davies et al. were performed in a bath gas of ~75% $N_2$ and 25% $O_2$ (with an ozonated oxygen flow added to ambient air).'

Page 3 line 3 – 4: '…observe a much lower sensitivity with ozone calibrations.' This is a particular strong statement, in light of the authors' results, to suggest the sensitivity is much lower. The statistically-significant lower sensitivity occurs only when $x_{O2} < 0.2$. Indeed, the $O_3$ calibration agrees near-exactly with that using $NO_2$ and aerosol-based calibrations when $x_{O2}$ approaches one.

Page 4 line 20: For the purposes of future work on calibrations by other researchers, further experimental details should be given here. Such details include the pressure and temperature of the sample in the PAS cell. Was the $NO_2$-laden $N_2$ pushed or pulled through the sample line? What was the total flow rate through the spectrometers? What was the 662 nm laser power, as measured by the photodiode?

Page 4 line 29: Again, further details should be given here. What powers did the lasers have, as measured by photodiode, during measurements? This quantity is very important given the photolysis observed by the authors. What was the pressure and temperature of the sample in the PAS cell? Was the $O_3$-laden sample pushed or pulled through the PAS and CRD? What was the total flow rate through the spectrometers?

Page 5 line 7: What is the particle cut diameter for a 0.57 millimetre orifice? Moreover, is 0.57 mm a diameter or radius?

Page 6 lines 11-13: Here, it is important to note that (i) the $O_3$ calibrations were only done at 532 and 662 nm, with the 406 nm PAS signal (which is of most relevance to the Bluvshtein study) showing, presumably, very little response to the $O_3$ concentrations generated. (ii) The calibration coefficient quoted is that for the fit through both the 532 and 662 nm calibration data.

Page 6 line 17: The sensitivity here is the average of 532, 662 and 780 nm data. It would be useful to have the calibrations for the individual wavelengths. Indeed, the 662 nm calibration in Figure 2 appears to agree with the sensitivity of the $NO_2$ calibration, while the 532 nm sensitivity is lower.

Page 6 line 28: 'it can be convenient to calibrate in air…'. Emphasise here that this is exactly the case for the Davies $et\ al.$ study. I think it is important to remind the reader of the Bluvhtein and Davies studies to put the current work in context.

Page 6 line 31: The reader is directed to Figure 4A. Why are there two series for 662 nm on this plot? Presumably, one of the series (the light red line) is actually for the 780 nm wavelength?

Page 7 line 11: The reader is now directed to Figure 4B. It is not clear what the different data series are. Please could the authors add a legend.

Page 9 line 2: The authors have now clearly demonstrated a significant impact of bath gas composition on PAS sensitivity. It would be useful at this point to put this result in context with their motivation for this study. The authors should again highlight the differences in $O_2$ composition for the Bluvshtein (10%) and Davies (25%) studies, and provide estimates of the approximate biases in calibrations that use these bath gas compositions. From the data here, it seems that the biases would be around 17% and 8% for $x_{O2}$ values of 0.1 and 0.25, respectively (assuming the PAS sensitivity behaviour at 405 nm is similar to the data measured here at longer visible wavelengths). Therefore, the current study into bath gas effects does not explain fully the factor of two discrepancy reported by Bluvshtein *et al*., but is expected to be a significant contributor.

Page 9 lines 6 – 7: 'excited state of the bath gas' should perhaps read 'excited state of the analyte'.

Page 10 line 2: 'as long as it is performed with 100% $O_2$ as the bath gas.' This statement seems unjustified as the authors data suggests that PAS sensitivity factors similar to that measured for $NO_2$ calibrations are found even at $x_{O2}$ values of ~0.17 when uncertainties in both the $O_3$ and $NO_2$ PAS sensitivities are considered. Certainly, though, the calibration coefficient markedly drops for $x_{O2}$ <0.1.

Page 10 lines 14 – 16: This sentence is not correct. Bluvshtein *et al*. use the values as currently quoted, but Davies *et al.* use an $O_3$ laden oxygen flow and add to air giving a PAS cell bath gas composition of approximately 25% $O_2$ and 75% $N_2$. These differences in bath gas composition should be stated clearly with estimates of the PAS calibration biases expected from the current study. Clearly, the current study does not completely reconcile the discrepancy.

**References**

1    N. Bluvshtein, J. M. Flores, Q. He, E. Segre, L. Segev, N. Hong, A. Donohue, J. N. Hilfiker and Y. Rudich, *Atmos. Meas. Tech.*, 2017, **10**, 1203–1213.

2    N. W. Davies, M. I. Cotterell, C. Fox, K. Szpek, J. M. Haywood and J. M. Langridge, *Atmos. Meas. Tech.*, 2018, **11**, 2313–2324.

---

## Referee Comment (RC2) · Anonymous Referee #2 · 28 Aug 2018

Review of "Technical Note: Can ozone be used to calibrate aerosol photoacoustic spectrometers?" by Fischer and Smith

Overall this paper is an important contribution to the literature on the use of ozone for calibration of photoacoustic instruments. A revision may include discussion of the following points.

1. Around line 25 on page 2, the important paragraph on previous work is presented. It would be good to go into the difference in the bath gas discussion between the Davies and the Bluvshtein work here, since this plays such a strong role in the results of the current paper.

[Figure]

2. The authors present amplitude measurements for photoacoustic signals from gases and aerosol, but no phase measurements. The phase information may help determine if time lags for physical/chemical processes are important.

3. Some of the historical literature on the subject may be useful, e.g., Harshbarger, W. R., and Robin, M. B. 1973. The opto-acoustic effect: Revival of an old technique for molecular spectroscopy. Acc. Chem. Res. 6:329–334. doi:10.1021/ar50070a001 for example.

4. Between lines 15 and 20 on page 4, use of copper tubing is mentioned for delivering NO2. We find that copper tubing can remove NO2 until passivated, though that's probably not important for the short length of tubing and the experiment here.

5. Around line 5 on page 5, " . . . particles were size selected at 500, 550, 600, and 650 nm using an electrostatic classifier .. " . Just to be sure, it would be good to specify if this is diameter or radius. Particle loss issues and thermal relaxation rate of aerosol might be important to study, and that distinction would be important.

6. In Fig. 4b an 'error bar' would be useful for the NO2 measurement.

7. Any speculations or discussion about why the NO2 calibration is higher than the nigrosine calibration?

8. The need to check calibration with various combinations of gases highlights the need for accurately measuring the resonance frequency and quality factor for quantitative measurements.
* * *

---

## Referee Comment (RC3) · Anonymous Referee #3 · 29 Aug 2018

The authors compare several calibrants for a photoacoustic spectrometer designed for the measurement of aerosol absorption, specifically nigrosine particles, $NO_2$, and $O_3$. Recently, two groups have reported contradictory results calibrating similar photoacoustic spectrometers with $O_3$. The photoacoustic spectrometer presented here uses several wavelengths to excite a single photoacoustic cell (where as previous reports were limited to a single wavelength per photoacoustic cell). Because the calibration is independent of excitation wavelength, calibrations at several wavelengths can be compared in addition to comparison between different calibrants. The authors also use a cold trap to introduce $O_3$ into the photoacoustic spectrometer in the absence of $O_2$. It is in the absence of $O_2$ that authors find dramatic differences in the calibration with $O_3$.

[Figure]

This paper is an important contribution to this ongoing discussion regarding calibration of photoacoustic spectrometers with O3 and is appropriate material for AMT. It is organized and well written. It should be published with a few minor changes.

Comments

1) I suspect that experiments in the absence of O2 are motivated by the reaction to reform O3 (O(3P) + O2 -> O3, JPL Publication 15-10, Chemical Kinetics and Photochemical Data for Use in Atmospheric Studies), however, this reaction is not discussed explicitly nor are the results compared with the known rate of this reaction. The authors should discuss this reaction, and it would be helpful if the authors attempted a simple box model of O3 photochemistry in the PAS cell.

2) In the discussion of photodissociation of O3 by 532 nm light, could the authors estimate the expected loss of O3 based on the residence time in the sample cell, O3 cross section, and assuming 100% yield to the photodisociation channel.

3) Page 2 line 26 – My understanding is that Bluvshtein et al. measured the RI only of the nigrosine independently. For other materials, the RI was retrieved from broadband extinction measurements.

4) Page 4 line 5-8 – Does an absorber need to be present to determine the resonant frequency? Or is a background signal used?

5) Figure 3 – Drifts in the PAS signal are not explained. Do the authors know the source of these drifts?

6) Figure 4a: The peach/orange color trace is mislabeled. I understand it to be 780 nm?

7) Figure 4b: In this figure it seems that the PAS sensitivity in the absence of O2 for 662 and 532 differ by 10-20%, but in figure 2 the calibration slope is nearly the same for 532 nm and 662 nm in N2. Why are they different?

[Figure]

8) Page 9 line 26: A scoping argument against a more through explanation. If not here than where? A simple boxmodel including the reaction to reform O3, could answer some of the questions raised here and on pg 6 line 28-29.

9) Page 10 line 14: missing negative in the sentence? I though the N2 bath gas was "inefficient"?

10) What is the O2 impurity in the N2 gas used? It may be useful to account for any trace O2 in these experiments.

---

## Author Comment (AC3) · 9 Oct 2018

1) I suspect that experiments in the absence of O2 are motivated by the reaction to reform O3 (O(3P) + O2 -> O3, JPL Publication 15-10, Chemical Kinetics and Photochemical Data for Use in Atmospheric Studies), however, this reaction is not discussed explicitly nor are the results compared with the known rate of this reaction. The authors should discuss this reaction, and it would be helpful if the authors attempted a simple box model of O3 photochemistry in the PAS cell.

The reviewer is correct, and the O(3P) + O2 -> O3 reaction is discussed on p. 7 line 6. We thank the reviewer for referring a citation for this reaction and have cited it in

the text. We performed a simple photodissociation calculation to assess the extent to which $O_3$ would be lost in our experiments (see point #2, below).

2) In the discussion of photodissociation of $O_3$ by 532 nm light, could the authors estimate the expected loss of $O_3$ based on the residence time in the sample cell, $O_3$ cross section, and assuming 100% yield to the photodisociation [sic] channel.

We thank the reviewer for this suggestion. We have carried out such a photodissociation calculation and estimate that most (much greater than 99.9%) of the $O_3$ should be dissociated. We surmise that the reason that only 5% of the $O_3$ was lost (Figure 3a) might result from rapid recombination to form $O_3$. We estimate that it would take only 4 ppm $O_2$ for recombination to compete with photodissociation. We have included this analysis in the discussion of Figure 3a.

3) Page 2 line 26 – My understanding is that Bluvshtein et al. measured the RI only of the nigrosine independently. For other materials, the RI was retrieved from broadband extinction measurements.

This is correct. Thank you for pointing out our error. The manuscript has been adjusted to reflect that only the RI of nigrosin was measured independently.

4) Does an absorber need to be present to determine the resonant frequency? Or is a background signal used?

Although an absorber can be used to determine the resonant frequency, it is not necessary and we conduct the frequency sweep in the bath gas without an absorber. Thank you for pointing out that we did not specify this; the manuscript has been updated to include this.

5) Figure 3 – Drifts in the PAS signal are not explained. Do the authors know the source of these drifts?

The drift in Figure 3B is likely due to drift in the $O_3$ concentration as the trap becomes depleted of $O_3$; we have included mention of this in the figure caption. The source
of drift in Figure 3A is not known, but it may reflect a drift in resonance frequency or temperature, but we point out that it does not affect the overall message of the figure: The effect of O3 photodissociation on the PAS signal is small and not enough to explain the factor of two discrepancy with and without O2 present. This analysis is now included in the body of the manuscript.

6) Figure 4a: The peach/orange color trace is mislabeled. I understand it to be 780 nm?

Thank you for pointing out our mistake. The legend has been updated to indicate the salmon-colored trace represents 780 nm.

7) Figure 4b: In this figure it seems that the PAS sensitivity in the absence of O2 for 662 and 532 differ by 10-20%, but in figure 2 the calibration slope is nearly the same for 532 nm and 662 nm in N2. Why are they different?

Figure 2 represents a single multi-point calibration, while Figure 4B represents multiple single-point calibrations. The latter case is especially susceptible to error as it represents a single measurement. Nonetheless, we do point out that the differences between the sensitivities for the three wavelengths is smaller with 100% O2 (7%) than with 100% N2 (17%), perhaps indicating that there is a real difference in the absence of O2. We do mention this possibility in the manuscript stating: "The differences in these values may reflect differences in the densities of states of the bath gas and the ozone when excited by the different wavelengths of light".

---

## Author Response (AR1)

**Responses to Referee #1**

*Page 1 line 18: The authors highlight the work of Bluvshtein et al. and Davies et al. in the abstract as partly motivating the current study. Therefore, the authors should summarise how the current work may reconcile these two studies, if at all, at the end of the abstract. As discussed further below, the authors incorrectly state in their conclusions that both the Bluvshtein et al. and Davies et al. studies use a bath gas of 10% O2 in N2; while this is the case for the Bluvshtein study, the Davies study is for ozone injected into ambient air (25% O2 in N2). These key differences in bath gas composition may partly account for calibration discrepancies between the two studies.*

**We appreciate this suggestion and have added a sentence at the end of the abstract briefly addressing this.**

Page 1 line 25: Some readers may not be familiar with a 'multipass enhancement cell'. The authors should make clear at this early stage what is meant by 'multipass' (i.e. multiple reflections of the excitation laser beam through the sample), although it is acknowledged that specific details of the multipass optical cell is given on page 3 line 11 – page 4 line 1.

**We have added a brief explanation of what a multipass enhancement cell is at this point**.

*Page 2 line 1: '...recent works exploring its validity have come to contradictory conclusions' should emphasise that these works were for visible laser wavelengths, i.e. '...recent works exploring its validity at visible wavelengths have come to contradictory conclusions'.*

**We made this change.**

*Page 2 line 2: '...discrepancy between ozone calibrations and particle-based calibrations'. The authors should state that this study was at a PAS laser wavelength of 405 nm only.*

**We made this change, also.**

*Page 2 line 16: 'Most commonly, this...'. 'This' is ambiguous and perhaps should read, 'Most commonly, this sensitivity factor...'.*

**We clarified this sentence by referencing the symbols used in Eq. 1.**

*Page 2 line 25: Replace '...light absorbing aerosols' with '...size selected light absorbing aerosols'.*

**We made this addition.**

*Page 2 line 27 - 28: Replace 'when they performed a calibration with ozone' with 'when they performed a calibration with ozone at a laser wavelength of 405 nm'.*

**We changed the sentence to reflect Bluvshtein et al. used a 405 nm laser.**

*Page 2 line 29: Add after 'Mie theory', 'at laser wavelengths of 405, 514 and 658 nm'.*

**This has been added at that point in the manuscript.**

*Page 2 line 32: The authors should state here: 'However, we note that the O3 calibrations performed by Bluvshtein et al. were in a bath gas composed of 90% N2 and 10% O2, while the calibrations of Davies et al. were performed in a bath gas of ~75% N2 and 25% O2 (with an ozonated oxygen flow added to ambient air).'*

**We thank the reviewer for making this suggestion, and have copied it into the manuscript.**

*Page 3 line 3 – 4: '...observe a much lower sensitivity with ozone calibrations.' This is a particular strong statement, in light of the authors' results, to suggest the sensitivity is much lower. The statistically-significant lower sensitivity occurs only when xO2 < 0.2. Indeed, the O3 calibration agrees near-exactly with that using NO2 and aerosol-based calibrations when xO2 approaches one.*

**We have changed this to read "…observe a lower sensitivity with ozone calibrations."**

*Page 4 line 20: For the purposes of future work on calibrations by other researchers, further experimental details should be given here. Such details include the pressure and temperature of the sample in the PAS cell. Was the NO2-laden N2 pushed or pulled through the sample line? What was the total flow rate through the spectrometers? What was the 662 nm laser power, as measured by the photodiode?*

**We appreciate this suggestion.  We have provided some more detail here.  See the comment under *Page 4 line 29* (below) for more information on the laser power.**

*Page 4 line 29: Again, further details should be given here. What powers did the lasers have, as measured by photodiode, during measurements? This quantity is very important given the photolysis observed by the authors. What was the pressure and temperature of the sample in the PAS cell? Was the O3-laden sample pushed or pulled through the PAS and CRD? What was the total flow rate through the spectrometers?*

**We appreciate this suggestion.  We have provided some more detail here, including details about T and P.  The incident beam powers have been listed under the instrument description. The multipass powers (those measured by the photodiode) are effective powers – they are relative, not absolute, as the PD calibration includes losses from windows and other optics.**

*Page 5 line 7: What is the particle cut diameter for a 0.57 millimetre orifice? Moreover, is 0.57 mm a diameter or radius?*

**Thanks for pointing out our typo.  It should have read "0.071 mm".  We have also included the cut point (~ 1100 nm) for the orifice used.**

*Page 6 lines 11-13: Here, it is important to note that (i) the O3 calibrations were only done at 532 and 662 nm, with the 406 nm PAS signal (which is of most relevance to the Bluvshtein study) showing, presumably, very little response to the O3 concentrations generated. (ii) The calibration coefficient quoted is that for the fit through both the 532 and 662 nm calibration data.*

**Actually, calibrations were performed at 532, 662, and 780 nm. We did not include 406 nm calibrations because of the low signal for that wavelength at the ozone concentrations used. We have added text here to clarify points (i) and (ii) and added text in the Ozone Calibration Methods section to clarify which wavelengths were used.**

*Page 6 line 17: The sensitivity here is the average of 532, 662 and 780 nm data. It would be useful to have the calibrations for the individual wavelengths. Indeed, the 662 nm calibration in Figure 2 appears to agree with the sensitivity of the NO2 calibration, while the 532 nm sensitivity is lower.*

**We have added the sensitivities derived from fitting the individual wavelengths as well for easier comparison. They differ by no more than 7%.**

*Page 6 line 28: 'it can be convenient to calibrate in air...'. Emphasise here that this is exactly the case for the Davies et al. study. I think it is important to remind the reader of the Bluvhtein [sic] and Davies studies to put the current work in context.*

**We added a reference to Davies et al. here.**

*Page 6 line 31: The reader is directed to Figure 4A. Why are there two series for 662 nm on this plot? Presumably, one of the series (the light red line) is actually for the 780 nm wavelength?*

**Thank you for pointing this out. The reviewer is correct, and we have corrected the plot accordingly.**

*Page 7 line 11: The reader is now directed to Figure 4B. It is not clear what the different data series are. Please could the authors add a legend.*

**We have added a legend to Figure 4B.**

*Page 9 line 2: The authors have now clearly demonstrated a significant impact of bath gas composition on PAS sensitivity. It would be useful at this point to put this result in context with their motivation for this study. The authors should again highlight the differences in O2 composition for the Bluvshtein (10%) and Davies (25%) studies, and provide estimates of the approximate biases in calibrations that use these bath gas compositions. From the data here, it seems that the biases would be around 17% and 8% for xO2 values of 0.1 and 0.25, respectively (assuming the PAS sensitivity behaviour at 405 nm is similar to the data measured here at longer visible wavelengths). Therefore, the current study into bath gas effects does not explain fully the factor of two discrepancy reported by Bluvshtein et al., but is expected to be a significant contributor.*

**We thank the reviewer for this helpful suggestion. We have added explicit discussion of how the findings from the current study might apply to the work of Bluvshtein et al. and Davies et al. Specifically, we point out that the O2 content of the bath gas does not seem to reconcile the discrepancy between these two studies.**

*Page 9 lines 6 – 7: 'excited state of the bath gas' should perhaps read 'excited state of the analyte'.*

**Actually, in this case we were in fact referring to the bath gas. However, this brings up a good point: Any long-lived excited state, whether it be the analyte, the bath gas, or an intermediate, would have a similar effect. We have added language to this effect here.**

*Page 10 line 2: 'as long as it is performed with 100% O2 as the bath gas.' This statement seems unjustified as the authors data suggests that PAS sensitivity factors similar to that measured for NO2 calibrations are found even at xO2 values of ~0.17 when uncertainties in both the O3 and NO2 PAS sensitivities are considered. Certainly, though, the calibration coefficient markedly drops for xO2 <0.1.*

**Yes, we agree with the reviewer. However, we note that (disregarding uncertainties) there is a slight upward trend in the data toward χ = 1. Although we cannot say whether this trend is "real" due to uncertainties, we prefer to err on the side of caution and recommend 100% O2 as the calibrant.**

*Page 10 lines 14 – 16: This sentence is not correct. Bluvshtein et al. use the values as currently quoted, but Davies et al. use an O3 laden oxygen flow and add to air giving a PAS cell bath gas composition of approximately 25% O2 and 75% N2. These differences in bath gas composition should be stated clearly with estimates of the PAS calibration biases expected from the current study. Clearly, the current study does not completely reconcile the discrepancy.*

**Thank you for pointing out our error. We've fixed this error and added estimates of the effect for each study.**

**Responses to Referee #2**

*1. Around line 25 on page 2, the important paragraph on previous work is presented. It would be good to go into the difference in the bath gas discussion between the Davies and the Bluvshtein work here, since this plays such a strong role in the results of the current paper.*

**We thank the reviewer for this suggestion and have added a couple of sentences discussing the differences at the end of that paragraph.**

*2. The authors present amplitude measurements for photoacoustic signals from gases and aerosol, but no phase measurements. The phase information may help determine if time lags for physical/chemical processes are important.*

**While we agree with the reviewer on this point, phase measurements were unfortunately not conducted due to limitations with the DAQ system and cannot be be presented here at present.**

*3. Some of the historical literature on the subject may be useful, e.g., Harshbarger, W. R., and Robin, M. B. 1973. The opto-acoustic effect: Revival of an old technique for molecular spectroscopy. Acc. Chem. Res. 6:329–334. doi:10.1021/ar50070a001 for example.*

**We thank the reviewer for making this suggestion and have referenced this paper in paragraphs 2 and 4 of the Introduction.**

*4. Between lines 15 and 20 on page 4, use of copper tubing is mentioned for delivering NO2. We find that copper tubing can remove NO2 until passivated, though that's probably not important for the short length of tubing and the experiment here.*

**We appreciate the referee bringing this to our attention. We observe the same problem (regardless of tubing composition), which is why we [1] keep the length of tubing short (~10cm) and [2] make an effort to passivate the tubing with a high concentration of NO2 prior to analysis. The manuscript has been updated here to reflect these things under "NO2 Measurements".**

*5. Around line 5 on page 5, " . . . particles were size selected at 500, 550, 600, and 650 nm using an electrostatic classifier .. " . Just to be sure, it would be good to specify if this is diameter or radius. Particle loss issues and thermal relaxation rate of aerosol might be important to study, and that distinction would be important.*

**Thank you for pointing out our imprecise language here. We have updated the manuscript to reflect that these values represent mobility diameters.**

6. In Fig. 4b an 'error bar' would be useful for the NO2 measurement.

**We thank the reviewer for the suggestion. We have added a 3σ error bar to the NO2 sensitivity in Figure 4B.**

*7. Any speculations or discussion about why the NO2 calibration is higher than the nigrosine calibration?*

**In figure 2, the NO2 sensitivity is 10% higher than the nigrosine sensitivity. We speculate that at least some of that difference could be attributed to uncertainty/error on CPC particle concentrations and in the refractive indices of nigrosin used in calculating the nigrosin absorbance. We have added a couple sentences to the results under "Non-ozone Methods of Calibration" indicating this.**

*8. The need to check calibration with various combinations of gases highlights the need for accurately measuring the resonance frequency and quality factor for quantitative measurements.*

**Yes, we agree! This is why we measure the resonant frequency each time the gas composition is changed or every 30 minutes, whichever comes first.**

**Responses to Referee #3**

*1) I suspect that experiments in the absence of O2 are motivated by the reaction to reform O3 (O(3P) + O2 -> O3, JPL Publication 15-10, Chemical Kinetics and Photochemical Data for Use in Atmospheric Studies), however, this reaction is not discussed explicitly nor are the results compared with the known rate of this reaction. The authors should discuss this reaction, and it would be helpful if the authors attempted a simple box model of O3 photochemistry in the PAS cell.*

**The reviewer is correct, and the O(3P) + O2 -> O3 reaction is discussed on p. 7 line 6. We thank the reviewer for referring a citation for this reaction and have cited it in the text. We performed a simple photodissociation calculation to assess the extent to which O3 would be lost in our experiments (see point #2, below).**

*2) In the discussion of photodissociation of O3 by 532 nm light, could the authors estimate the expected loss of O3 based on the residence time in the sample cell, O3 cross section, and assuming 100% yield to the photodisociation [sic] channel.*

**We thank the reviewer for this suggestion. We have carried out such a photodissociation calculation and estimate that most (much greater than 99.9%) of the O3 should be dissociated. We surmise that the reason that only 5% of the O3 was lost (Figure 3a) might result from rapid recombination to form O3. We estimate that it would take only 4 ppm O2 for recombination to compete with photodissociation. We have included this analysis in the discussion of Figure 3a.**

*3) Page 2 line 26 – My understanding is that Bluvshtein et al. measured the RI only of the nigrosine independently. For other materials, the RI was retrieved from broadband extinction measurements.*

**This is correct. Thank you for pointing out our error. The manuscript has been adjusted to reflect that only the RI of nigrosin was measured independently.**

*4) Does an absorber need to be present to determine the resonant frequency? Or is a background signal used?*

**Although an absorber can be used to determine the resonant frequency, it is not necessary and we conduct the frequency sweep in the bath gas without an absorber. Thank you for pointing out that we did not specify this; the manuscript has been updated to include this.**

*5) Figure 3 – Drifts in the PAS signal are not explained. Do the authors know the source of these drifts?*

**The drift in Figure 3B is likely due to drift in the O3 concentration as the trap becomes depleted of O3; we have included mention of this in the figure caption. The source of drift in Figure 3A is not known, but it may reflect a drift in resonance frequency or temperature, but we point out that it does not affect the overall message of the figure: The effect of O3 photodissociation on the PAS signal is small and not enough to explain the factor of two discrepancy with and without O2 present. This analysis is now included in the body of the manuscript.**

*6) Figure 4a: The peach/orange color trace is mislabeled. I understand it to be 780 nm?*

**Thank you for pointing out our mistake. The legend has been updated to indicate the salmon-colored trace represents 780 nm.**

*7) Figure 4b: In this figure it seems that the PAS sensitivity in the absence of O2 for 662 and 532 differ by 10-20%, but in figure 2 the calibration slope is nearly the same for 532 nm and 662 nm in N2. Why are they different?*

**Figure 2 represents a single multi-point calibration, while Figure 4B represents multiple single-point calibrations. The latter case is especially susceptible to error as it represents a single measurement. Nonetheless, we do point out that the differences between the sensitivities for the three wavelengths is smaller with 100% O2 (7%) than with 100% N2 (17%), perhaps indicating that there is a real difference in the absence of O2. We do mention this possibility in the manuscript stating: "
[revised manuscript text omitted]

J. B. Burkholder, S.P. Sander, J. A. J. . B. R. H. C. K. M. K. V. O. D. W. and Wine, P.: Chemical Kinetics and Photochemical Data for Use in Atmospheric Studies, Evaluation No. 18, JPL Publication 15-10, https://jpldataeval.jpl.nasa.gov/, 2015.

Japar, S. M. and Szkarlat, A. C.: Measurement of diesel vehicle exhaust particulate using photoacoustic spectroscopy, Combust. Sci. Technol., 24, 215–219, https://doi.org/10.1080/00102208008952440, 1980.

Kalkman, J. and van Kesteren, H.: Relaxation effects and high sensitivity photoacoustic detection of NO2 with a blue laser diode, Appl. Phys. B: Lasers Opt., 90, 197–200, https://doi.org/10.1007/s00340-007-2895-0, 2008.

Kosterev, A., Mosely, T., and Tittel, F.: Impact of humidity on quartz-enhanced photoacoustic spectroscopy based detection of HCN, Appl. Phys. B: Lasers Opt., 85, 295–300, https://doi.org/10.1007/s00340-006-2355-2, 2006.

Lack, D. A., Lovejoy, E. R., Baynard, T., Pettersson, A., and Ravishankara, A. R.: Aerosol absorption measurement using photoacoustic spectroscopy: Sensitivity, calibration, and uncertainty developments, Aerosol Sci. Technol., 40, 697–708, https://doi.org/10.1080/02786820600803917, 2006.

Lack, D. A., Richardson, M. S., Law, D., Langridge, J. M., Cappa, C. D., McLaughlin, R. J., and Murphy, D. M.: Aircraft instrument for comprehensive characterization of aerosol optical properties, Part 2: Black and brown carbon absorption and absorption enhancement measured with photo acoustic spectroscopy, Aerosol Sci. Technol., 46, 555–568, https://doi.org/10.1080/02786826.2011.645955, 2012.

Lambe, A. T., Cappa, C. D., Massoli, P., Onasch, T. B., Forestieri, S. D., Martin, A. T., Cummings, M. J., Croasdale, D. R., Brune, W. H., Worsnop, D. R., and Davidovits, P.: Relationship between oxidation level and optical properties of secondary organic aerosol, Environ. Sci. Technol., 47, 6349–6357, https://doi.org/10.1021/es401043j, 2013.

Lewis, K., Arnott, W. P., Moosmüller, H., and Wold, C. E.: Strong spectral variation of biomass smoke light absorption and single scattering albedo observed with a novel dual-wavelength photoacoustic instrument, J. Geophys. Res.: Atmos., 113, https://doi.org/10.1029/2007jd009699, 2008.

Miklós, A., Hess, P., and Bozóki, Z.: Application of acoustic resonators in photoacoustic trace gas analysis and metrology, Rev. Sci. Instrum., 72, 1937–1955, https://doi.org/10.1063/1.1353198, 2001.

Moosmüller, H., Arnott, W. P., Rogers, C. F., Chow, J. C., Frazier, C. A., Sherman, L. E., and Dietrich, D. L.: Photoacoustic and filter measurements related to aerosol light absorption during the Northern Front Range Air Quality Study (Colorado 1996/1997), J. Geophys. Res.: Atmos., 103, 28 149–28 157, https://doi.org/10.1029/98jd02618, 1998.

Roessler, D. M. and Faxvog, F. R.: Photoacoustic determination of optical absorption to extinction ratio in aerosols, Appl. Opt., 19, 578, https://doi.org/10.1364/ao.19.000578, 1980.

Silver, J. A.: Simple dense-pattern optical multipass cells, Appl. Opt., 44, 6545, https://doi.org/10.1364/ao.44.006545, 2005.

Wiegand, J. R., Mathews, L. D., and Smith, G. D.: A UV–vis photoacoustic spectrophotometer, Anal. Chem., 86, 6049–6056, https://doi.org/10.1021/ac501196u, 2014.

Yung, Y. and DeMore, W.: Photochemistry of Planetary Atmospheres, Oxford University Press, New York, New York, 1999.

Zhang, X., Kim, H., Parworth, C. L., Young, D. E., Zhang, Q., Metcalf, A. R., and Cappa, C. D.: Optical properties of wintertime aerosols from residential wood burning in Fresno, CA: Results from DISCOVER-AQ 2013, Environ. Sci. Technol., 50, 1681–1690, https://doi.org/10.1021/acs.est.5b04134, 2016.